# Lipotoxicity as a Barrier for T Cell-Based Therapies

**DOI:** 10.3390/biom12091182

**Published:** 2022-08-25

**Authors:** Romy Böttcher-Loschinski, Judit Rial Saborido, Martin Böttcher, Sascha Kahlfuss, Dimitrios Mougiakakos

**Affiliations:** 1Department of Hematology and Oncology, University Hospital Magdeburg, Otto-von-Guericke University Magdeburg, 39120 Magdeburg, Germany; 2Medical Department 5–Hematology and Oncology, University Hospital Erlangen, Friedrich-Alexander-University of Erlangen-Nürnberg, 91054 Erlangen, Germany; 3Health Campus Immunology, Infectiology, and Inflammation (GCI3), Medical Center, Otto-von-Guericke University Magdeburg, 39120 Magdeburg, Germany; 4Institute of Molecular and Clinical Immunology, Medical Faculty, Otto-von-Guericke University Magdeburg, 39120 Magdeburg, Germany; 5Institute of Medical Microbiology and Hospital Hygiene, Medical Faculty, Otto-von-Guericke University Magdeburg, 39120 Magdeburg, Germany; 6CHaMP, Center for Health and Medical Prevention, Otto-von-Guericke-University Magdeburg, 39120 Magdeburg, Germany

**Keywords:** lipotoxicity, T cell, immunometabolism, tumor metabolism, tumor microenvironment, cell-based therapy, immunotherapy

## Abstract

Nowadays, T-cell-based approaches play an increasing role in cancer treatment. In particular, the use of (genetically engineered) T-cells has heralded a novel era for various diseases with previously poor outcomes. Concurrently, the relationship between the functional behavior of immune cells and their metabolic state, known as immunometabolism, has been found to be an important determinant for the success of immunotherapy. In this context, immune cell metabolism is not only controlled by the expression of transcription factors, enzymes and transport proteins but also by nutrient availability and the presence of intermediate metabolites. The lack of as well as an oversupply of nutrients can be detrimental and lead to cellular dysfunction and damage, potentially resulting in reduced metabolic fitness and/or cell death. This review focusses on the detrimental effects of excessive exposure of T cells to fatty acids, known as lipotoxicity, in the context of an altered lipid tumor microenvironment. Furthermore, implications of T cell-related lipotoxicity for immunotherapy will be discussed, as well as potential therapeutic approaches.

## 1. Introduction

T cells have an impressive ability to detect and eliminate tumors in an antigen-specific manner. The latter feature is also one of the main reasons why T cells have come into focus for (targeted) immune cell-based therapies in cancer. Here, the adoptive transfer of autologous or allogeneic T cells, which recognize and attack cancer cells, has been an especially effective concept for several malignancies [1]. Current adoptive T cell-based therapies include ex vivo-expanded autologous tumor-infiltrating T cells (TILs), donor lymphocyte infusions (DLIs) after allogeneic hematopoietic stem cell transplantation (alloHSCT), and genetically modified T-lymphocytes carrying chimeric antigen receptors (CARs) or transgenic T cell receptors (TCRs) (Figure 1). Despite all of these amazing advances, T cell-based therapies fail in a substantial proportion of patients. This may be due to loss of the target antigen or other inhibitory factors present within the tumor environment (TME) [2]. These inhibitory mechanisms include direct immunosuppression by immune checkpoints and a T cell-inhospitable TME, leading, among other things, to reduced persistence and a loss of effector functions [3].

The TME is a unique tumor-supporting niche consisting of cellular components, such as myeloid-derived suppressor cells (MDSCs), tumor-associated macrophages (TAMs), cancer-associated fibroblasts, and non-cellular components, e.g., soluble factors including bioenergetic substrates and bioactive metabolites. Furthermore, depletion of critical nutrients and simultaneous abundance of alternative nutrients and metabolic byproducts can promote metabolic reprogramming of tumor or immune cells or lead to dysfunction and cell death. In particular, solid tumors are known to have a nutrient-poor TME, limiting the infiltration and efficacy of adoptively transferred T cells [1]. Prominently, glucose is depleted by glycolytic tumor cells of various entities, resulting in dysfunctional TILs and tumor progression [4]. Accumulation of lactate, a byproduct of glucose utilization by enhanced aerobic glycolysis (in malignant cells), further abolishes T cell proliferation, activation, and function [5]. In the last decade, studies have begun to investigate nutrients beyond glucose within the TME. For instance, the TME has been shown to be enriched in energy-rich lipids, which promote cancer progression in various tumor entities [6].

Lipids are a heterogeneous group of metabolites, which mainly consist of fatty acids (FAs), cholesterol, and phospholipids. They are important for building cell and organelle membranes, providing cellular energy, and are involved in intracellular and hormonal signaling. Under homeostatic conditions, T cells also rely on FAs to some extent, depending on their developmental and differential status [7]. However, it has been described that excessive uptake and storage of fatty acids from their environment leads to T cell dysfunction and cell death [8]. These detrimental effects of excessive lipid storage in non-adipose tissue leading to cellular damage and cell death are defined as lipotoxicity. Thus far, lipotoxicity is mainly described in the setting of obesity and focuses on damage occurring in the liver, kidneys, pancreas, heart, and skeletal muscles [9,10]. However, lipotoxicity also occurs in the tumor-related setting and impairs the anti-tumor responses of immune cells [6,11]. Thus, it is important to understand the consequences of high lipid contents in the TME and to find strategies to overcome this metabolic barrier for efficient tumor therapy.

This review provides an overview of the altered lipid TME (co-)caused by obesity and age and the (underlying) reciprocal reprogramming of tumor cells and adipocytes. Furthermore, the mechanisms of lipotoxicity and the implications for tumor-related T cells and T cell-based therapies are discussed. Finally, potential therapeutic approaches to prevent T cell lipotoxicity will be outlined.

## 2. Sources of Lipids within the Tumor Microenvironment

### 2.1. Risk Factors Obesity and Age

Obesity is acknowledged as a major risk factor for the development of several cancer types [12,13]. Especially, malignancies of female hormone-sensitive tissues and of the digestive tract are linked to excessive body weight [14]. However, an increasing body of research also suggests an elevated relative risk in other cancer entities, such as hematologic malignancies [15,16,17]. Additionally, the development of overweight and obesity during childhood and early adulthood seems to further increase the risk of developing pancreatic cancer, colon cancer, and multiple myeloma [18,19,20]. Strikingly, the prevalence of obesity has been constantly increasing within the last decades, leading to about a third of the global population being obese or overweight [21]. Thus, understanding the connection between cancer and obesity is crucial for tumor prevention and prospective cancer therapies.

Obesity results from a combination of physical inactivity, uptake of excess calories, and genetic background. The systemic dysregulation in obesity is complex and ranges from dyslipidemia and insulin insensitivity to chronic low-grade inflammation and hormonal dysbalances. All these imbalances can have an impact on tumor and immune cells. For instance, T cell proliferation and counts are decreased, and the T cell subset composition is altered in obese as compared to lean individuals in non-cancer-related settings [22,23,24,25,26]. For instance, Wijingaarden et al. reported a shift from naïve T cells and effector memory cells re-expressing CD45RA T cells (T_EMRA_) towards a higher proportion of central-memory T cells (T_CM_) for CD4^+^ Tells in morbidly obese patients compared to lean controls. Contrarily, these patients also showed a higher proportion of CD8^+^ naïve T cells, effector memory T cells (T_EM_) and T_EMRA_ as well as a decreased T_CM_ compartment. Of note, cytokine production of IL-2 and IFN-γ was decreased in CD8^+^ but not CD4^+^ T cells in those patients compared to lean controls [26]. Obesity affects T cell formation already in the thymus, resulting in accelerated thymic aging and a limited T cell repertoire [27]. Lymphatic and adipose tissue T cells were reported to have an exhausted phenotype in obese humans and mice [28,29]. Furthermore, it was recently published that high-fat diet-induced obesity in mice leads to metabolic reprogramming of tumor cells and the TME, which in turn results in CD8^+^ T cell exhaustion and failure of tumor control by T cells [13,30].

Cancer is mainly seen as a disease of the elderly. Interestingly, aging and obesity share similar conditions and comorbidities, such as low-grade chronic inflammation, immune cell exhaustion and senescence [31,32,33,34,35] (Figure 2). Dyslipidemia, expansion of visceral adipose tissue (VAT), a type of white adipose tissue, and elevation of body fat percentages are common in aged and obese individuals [36,37,38,39,40]. Likewise, the composition of the bone marrow shifts by preferential differentiation from mesenchymal stem cells (MSCs) to adipocytes, resulting in an increase of marrow adipose tissue (MAT) during aging [41,42]. Increased MAT is also observed in obese individuals [43]. Importantly, adipocytes are one of the main cellular compartments in the TME in several tumor types and have been shown to promote tumor cell survival and proliferation and to affect effector functions of immune cells, such as T cells, NK cells, macrophages and dendritic cells [44,45].

In conclusion, obesity and aging both lead to systemic alterations that ultimately increase the oncogenic risk and impair immune cell function (Figure 2). However, how the systemic metabolic dysregulation in obesity and aging promotes metabolic changes and function in tumor cells, immune cells, and the TME is not fully understood yet and needs to be further elucidated. Furthermore, tumor cells themselves can also shape their own lipid-enriched environment. The following chapter will provide a more detailed overview of the role of lipids in tumor cells and the TME.

### 2.2. Lipids Support Cancer Cell Survival and Progression

Tumor cells are shaped by their surroundings and, vice versa, shape their own (metabolic) environment. The recently published “hallmarks of cancer” include sustained proliferative signaling and replicative immortality [47]. Paired with a deregulated cellular metabolism, the need for nutrients fueling excessive proliferation and survival of tumor cells cumulates leads to glucose depletion and hypoxia. In fact, usage of lipids could therefore display an alternative bioenergetic source.

In solid tumors, such as breast cancer, lung cancer, glioma, and hepatocellular carcinoma, hypoxia and glucose depletion promote lipoprotein lipase (LPL) in tumor and stromal cells, which is important to release free fatty acids (FFAs) from extracellular triglycerides carried in low-density lipoproteins (LDL). Furthermore, tumor cells activate TME-associated adipocytes, which then induce lipolysis of stored triglycerides (TGs) and secrete FAs. These are then mainly taken up by tumor cells via fatty acid transporters, such as CD36 and FABP4 [48,49,50,51,52,53]. Simultaneous upregulation of autophagy supporting lipid degradation and FA release as well as mitochondrial fatty acid oxidation (FAO) allows for overcoming glucose deprivation and promotes tumor growth and survival [54]. Interestingly, elevated expression of genes related to lipid metabolism has been proposed as poor prognostic biomarkers for several cancer entities [55].

Similar reprogramming can also be found in hematologic malignancies. Leukemic cells and leukemic stem cells reside in the bone marrow and lymphatic niches that resemble a glucose-deprived and hypoxic TME, which is comparable to the conditions seen in solid tumors. Two recent studies demonstrated that acute myeloid leukemia (AML) cells remodel their direct environment by secretion of exosomes that, in turn, promote MSC differentiation towards adipocytes and lipolysis in already differentiated adipocytes [56,57]. Moreover, adipose tissue is beneficial for AML blasts and leukemic stem cells (LSCs) by increasing FAO and CD36/FABP4 expression, resulting in the prevention of serum starvation-induced apoptosis and protection from chemotherapy [58,59]. In fact, an enhanced FAO rate is accompanied by resistance to chemotherapeutic drugs in AML [59,60,61]. B cell chronic lymphocytic leukemia (CLL) cells also display a higher need for cholesterol and FA compared to regular mature B cells [62,63]. Interestingly, the majority of CLL patients display elevated levels of LDL in the blood and have a higher incidence of dyslipidemia at diagnosis [64,65]. The observed hypercholesterolemia could result from malnutrition. However, it is also discussed as a paraneoplastic syndrome because CLL cells produce oleoethanolamide, leading to the release of FFAs by adipocytes into the TME. In turn, FFAs in the blood plasma can then be packed into LDL in the liver [63,66]. As mentioned before, the breakdown of LDL is facilitated by LPL, amongst others, generating ligands for the nuclear receptor peroxisome proliferator-activated receptor (PPAR)α. Activation of PPARα in CLL cells leads to increased FAO, expression of the immunosuppressive IL-10, and better survival [67]. High expression of LPL and PPARα in CLL have been proposed as possible prognostic markers for poor prognosis in several studies [67,68,69,70]. Furthermore, lysosomal degradation by lysosomal lipase of LDLs into FA, cholesterol, and vitamin E promotes growth and survival signals in activated CLL cells [63]. Interestingly, McCaw et al. also state that slow-growing CLL cells, but not rapidly growing CLL cells, depend on the breakdown of LDL [63]. This is controversial to the high LPL and PPARα expression in aggressive (and highly proliferative) CLL cells and raises the question of what role LDLs and fatty acids might play during different CLL disease stages, therapy-evading LSCs, and their TME.

Cancer cells have also been described to increase de novo fatty acid synthesis (FAS) to fuel their excessive need for nutrients. Using intermediates of glycolysis and the tricarboxylic acid (TCA) cycle for FAS further demonstrates the high flexibility of cancer cells to adapt their metabolism to meet their need for FAs as substrates for anabolic processes and structural components as well as to react to the reduced availability of FAs in the TME [71,72]. For instance, poor-prognosis primary effusion lymphoma (PEL) and other B cell non-Hodgkin lymphomas (B-NHLs) exhibit high levels of aerobic glycolysis, providing substrates for elevated FAS. It is discussed that the newly synthesized FAs are then incorporated into the cell membrane, supporting the high proliferative capacity of these cells [73]. Furthermore, inhibition of the fatty acid synthetase (FASN), a key molecule for FAS, in DLBCL, a form of B-NHLs, induces apoptosis. Here, the inhibition of FASN is discussed to result in a lower phospholipid production, which changes the tumor cell membrane composition and decreases c-Met membrane location and signaling. Consequently, suppressed c-Met signaling leads to the impaired downstream activity of the PI3K/AKT pathway and inhibits cell proliferation and apoptosis in these DLBCL cells [74]. Additionally, overexpression of FASN is not only involved in lipid synthesis but also in driving proliferation via extracellular regulated kinase 1/2 (ERK 1/2) and phosphatidylinositol-3′-kinase (PI3K/AKT), autophagy, DNA repair, lysosome biogenesis, cytoskeletal remodeling and metastasis [75]. Therefore, it is believed that de novo synthesized FAs are not released into the TME and are rather used for intracellular processes [71,72].

Conclusively, a lipid-enriched TME can be found in many malignant entities. Adipocytes are abundantly found in the TME. They have been shown to promote tumor progression and affect immune cell effector functions by TG storage, the release of FFAs, and the secretion of hormones, cytokines and growth factors. Vice versa, tumor cells themselves reprogram adipocytes to meet their needs (Figure 3) [44,45]. However, the interplay of tumor and neighboring cells, as well as the network of lipid metabolic pathways in these cells, is complex. Further research is needed to understand the role of lipids in tumor cells and the TME, identify overlapping lipid metabolic alterations in tumors, and translate these as targets into the clinic.

## 3. Mechanisms of Cellular Lipotoxicity

As described above, the presence of pathological levels of lipids can occur at both the systemic level and in the TME. Tumor cells upregulate proteins to use these lipids. However, cells that do not display the same deregulated metabolism (as tumor cells) might experience cytotoxic effects from the overabundance of lipids. In this context, lipotoxicity can manifest itself via a variety of mechanisms, which are summarized in Figure 4. These include oxidative stress and mitochondrial dysfunctions, endoplasmic reticulum stress, ferroptosis, and autophagy, which will each be described in detail in the following sections.

### 3.1. Oxidative Stress and Mitochondrial Dysfunction

Oxidative stress is a metabolic condition describing the accumulation of reactive oxygen species (ROS). The superoxide anion radical (O_2_^●−^) and hydrogen peroxide (H_2_O_2_) are the major ROS produced by a range of enzymes, such as NAPDH oxidases (NOX) and the mitochondrial electron transport chain [76]. One of the key components that links FFA and oxidative stress is the fatty acid transporter CD36. This protein is expressed in most cell types, and several studies in human tissues have shown increased expression of CD36 followed by increased lipid uptake, ROS formation, and apoptosis [77]. It is worth mentioning that CD36 not only facilitates the uptake of FFAs but also acts as a signal-transducing receptor for oxidized low-density lipoproteins (oxLDL) and danger-associated or pathogen-associated molecular patterns (DAMPs and PAMPs), leading to the activation of transcription factors such as PPARγ and NF-κB [78]. Once FFAs are taken up into the cell, they can be used for β-oxidation and energy production, stored in lipid droplets for later use or accumulated in the cytosol. In the latter situation, an excess of non-stored FFAs leads to increased oxidative stress and results in lipotoxicity. Given that mitochondria are the major source of ROS, any alterations in mitochondrial metabolism and function might be harmful. Several mechanisms have been proposed to explain how FFAs impair mitochondria. Szeto et al. suggested a direct impact of FFAs in mitochondrial morphology, leading to loss of mitochondrial christae and, consequently, to disruptions in the electron transport chain (ETC) in renal cells [79]. Furthermore, unsaturated fatty acids disrupt the proton gradient by increasing proton conductance at the inner mitochondrial membrane resulting in the formation of the permeability transition pore (PTP) and the release of mitochondrial apoptogenic proteins into the cytosol [80]. Mitochondrial dysfunction might also be caused by progressively increased damage of mitochondrial DNA (mtDNA), proteins, and lipids caused by cytosolic and mitochondrial ROS, which is the result of exposure to increasing concentrations of FFAs [81,82]. For instance, Dludla et al. reported reduced levels of ubiquinone, a part of the electron transport chain, and consequently increased mitochondrial ROS production in H9c2 cardiomyoblasts when overloaded with palmitic acid [83]. FFAs further induce ceramide accumulation, which has been shown to result in mitochondrial protein hyperacetylation and subsequently mitochondrial dysfunction [84]. Conclusively, a surplus of fatty acids results in mitochondrial dysfunction and increased ROS production, which is the basis for the other mechanisms of lipotoxicity.

### 3.2. Endoplasmatic Reticulum Stress

In a study, exposure of podocytes to palmitic acid led to increased ROS levels, followed by Ca^2+^ release from the endoplasmatic reticulum (ER), Ca^2+^ depletion and finally, ER stress, which resulted in apoptosis [85]. The ER has many important functions, such as protein folding and maturation by the addition of post-translational modifications as well as lipid synthesis and the regulation of calcium homeostasis. Defects in these processes, as well as oxidative stress, lead to the accumulation of misfolded or aggregated proteins, thereby activating the unfolded protein response (UPR). Its main function is to restore ER homeostasis and function by orchestrating the degradation of misfolded proteins. Despite its essential role in preventing ER stress, an excessive UPR activation may result in cell death [86,87].

Several studies have shown that an excess of FFAs induces ER stress and inflammation in different tissues, including the liver, adipose tissue [88], and hepatocytes [89]. FFAs can activate UPR indirectly and directly. Differences in lipid composition could cause changes in protein folding or trafficking through the ER, thereby resulting in increased unfolded proteins and activation of the UPR pathway [90]. Disruption of ER calcium homeostasis by lipid-mediated inhibition of the sarco/endoplasmic reticulum Ca^2+^-ATPase (SERCA) is also possible and has been shown for hepatocytes of obese mice and murine cholesterol-loaded macrophages [91,92]. Besides that, lipids can directly induce UPR.

The UPR pathway is mainly activated by three transmembrane proteins that act as unfolded protein sensors: inositol requiring enzyme 1 (IRE1α), transcription factor activating transcription factor 6 (ATF6), and protein kinase RNA-like endoplasmic reticulum kinase (PERK) [87]. The three main components of the UPR activate independent signaling pathways. The most studied protein, given that it is present in yeast, is IRE1α. IRE1α is a bifunctional protein with both a kinase and endoribonuclease function. Once IRE1α is oligomerized in the ER membrane, its endoribonuclease function is activated and cleaves the mRNA-encoding X-box binding protein 1 (XBP1) at two specific positions. The resulting active form of XBP1 regulates lipid biosynthetic enzymes and ER-associated degradation components [93]. Although this pathway is supposed to alleviate ER stress, in the context of lipotoxicity, several studies have shown that FFAs hold the potential to induce ER stress through IRE1α activation. As an example, IRE1α signaling plays a role in FA-induced cell death of pancreatic β-cells [94,95]. The second UPR component is ATF6, which upon accumulation of unfolded proteins, is packed and transported to the Golgi apparatus. Once there, it is cleaved by two proteases, and the N-terminal fragment is released, which then enters the nucleus to activate UPR target genes [96,97]. However, the mechanisms whereby ATF6 responds to ER stress are still not well elucidated. Lastly, upon activation, PERK phosphorylates itself and the translation initiation factor eIF2α, resulting in mRNA translation inhibition. This culminates in a reduction of protein synthesis and therefore alleviates ER stress [87]. Similar to IRE1α activation, FFAs seem to be able to activate PERK [95,98]. However, PERK activation does not seem to be sufficient to prevent FFA-induced apoptosis [99]. PERK-mediated apoptosis is facilitated by the phosphorylation of eIF2α, which induces the translation of activating transcription factor 4 (ATF4). ATF4, in turn, binds to several proteins, including CHOP (DDIT3, DNA damage-inducible transcript 3) and DNA damage-inducible protein (GADD34). CHOP controls the expression of apoptosis-involved genes. Both ATF4 and CHOP mRNA and protein levels were increased in human hepatocytes [100,101] and neuroblastoma cells [102] exposed to high levels of FFAs, suggesting that FFAs may induce ATF4 expression. Interestingly, IRE1 and PERK are active as a dimer/oligomeric state, which is supported by the biophysical property of the ER membrane. Changes in lipid composition and thereby of the biophysical property of the ER membrane by loading murine cells with palmitate has been shown to activate both complexes independent of unfolded proteins and to increase the response to unfolded proteins [103].

Overall, oxidative stress caused by excessive FFAs results in oxidized misfolded proteins that activate the UPR machinery in the ER. However, the overload of misfolded proteins and the excessive activation of this pathway leads to ER stress, which can be detrimental to the cell, leading to cell death.

### 3.3. Ferroptosis

In recent years, a new mechanism of regulated cell death, in which lipids play a primary role, has been described as ferroptosis. Ferroptosis is an iron-dependent form of programmed cell death induced by oxidative stress and lipid peroxidation [104]. Lipid peroxidation is a process that results from oxidative stress, where oxidants, free radicals and non-radical species attack the carbon-carbon double bond(s) from lipids, primarily in polyunsaturated fatty acids (PUFAs), resulting in lipid hydroperoxides (LOOH) and peroxyl radicals [105]. In detail, ferroptosis can be caused by intracellular glutathione (GSH) depletion and impaired glutathione peroxidase 4 (GPX4) activity. This, in combination with damaged lipids resulting from Fe^2+/3+^-mediated peroxidation, known as the Fenton reaction, leads to ROS accumulation and, subsequently, ferroptosis. In addition to the Fenton reaction, iron may increase the activity of enzymes responsible for lipid peroxidation and oxygen homeostasis, such as lipoxygenase (ALOX) or prolyl hydroxylases (PHD) [106,107].

Besides these biochemical features, ferroptosis can be defined by cellular morphology. Cells undergoing ferroptosis usually display an incomplete plasma membrane, mitochondrial shrinkage with increased membrane density, and diminished or absent cristae [106,107,108]. Regarding genetic characteristics, several genes and proteins have been defined as biomarkers for ferroptosis. These include prostaglandin-endoperoxide synthase 2 (PTGS2/COX2), responsible for prostaglandin biosynthesis [109], and Acyl-CoA synthetase long-chain family members (ACSL), involved in fatty acid metabolism [110,111,112]. The latter drives ferroptosis by increasing the PUFA content in phospholipids, which makes them more susceptible to lipid peroxidation and, subsequently, ferroptosis.

### 3.4. Autophagy and Lysosomal Dysfunction

Autophagy is the process by which malfunctional organelles and cellular components, such as oxidized proteins and lipids, are degraded in a process of self-digestion that takes place in the lysosomes [113]. It is essential to regulate cell homeostasis, and it is a potent tool to alleviate FFA-mediated lipotoxic effects. Several studies show that treatment with increasing concentrations of FAs helps overcome FFA-induced apoptosis by induction of autophagy [114,115]. However, the mechanism behind this FA-mediated autophagy induction is not yet fully understood. Previous observations suggest an involvement of c-Jun N-terminal kinases (JNKs) [114,116,117], whereas others show that a reduction in the AKT/mTOR signaling pathway increases autophagy independent of JNKs [115]. Besides, ceramides also have the potential to trigger autophagy, which can be perceived as an indirect FFA-elicited effect [118]. The prolonged overload of lipids, oxidized proteins, and nonfunctional organelles such as mitochondria leads to constitutive activation of autophagy, resulting in lysosomal dysfunction, which, in turn, limits autophagy and leads to cell death [119].

The relationship between oxidative stress, ER stress, and autophagy is still not fully clear. One study showed that autophagy is a prerequisite for UPR activation in palmitic acid-treated β-cells [120]. Similarly, as mentioned above, autophagy can be the result of a direct FFA-induced JNK activation, leading to ER stress [116]. In addition, ROS generation can result from FFA-induced autophagy that activates the protein kinase C α (PKCα)/NOX4pathway [121]. As mentioned above, NOX are a source of ROS by catalyzing the transfer of electrons from NADPH to O_2_, resulting in superoxide formation [122]. All of these studies point to a relationship between these processes that is not yet fully understood. Alternatively, autophagy can also take place independently of ROS and ER stress [117]. Despite these documented autophagy-promoting roles, other publications claim the opposite. Actually, several studies show that FFAs can inhibit autophagy, which worsens their detrimental impact. Therefore, whether FFAs have an autophagy-promoting or inhibiting role and how ROS, ER stress and autophagy are interconnected still needs to be deciphered [123,124,125].

In summary, lipotoxicity is associated with oxidative stress in the cells, and its mechanisms are closely intertwined. However, responses to excess FFAs can be cell type- and TME-dependent and should therefore be further explored. Since the TME can be enriched with FFAs, it is important to know how tumor, stromal and immune cells react to a high lipid burden to find targets for improving cancer therapy.

## 4. Implications of Lipotoxicity on T Cell-Based Therapies

T cells are one of the major immune subsets for tumor control. Therefore, T cells have been extensively studied and exploited for cell-based therapies. However, depending on the T cell subset, high lipid content in their environment is detrimental to their effector functions and survival. 

### 4.1. Differential Effects of FAs in T Cell Subsets

It is well known that T cells adapt their metabolism during the different stages of T cell activation and differentiation [126]. In addition, microenvironmental cues, such as the lack or abundance of certain nutrients, can also have an impact on T cell metabolic activity. For instance, several studies have focused on the effects that lipids, particularly fatty acids, might have on T cells. In general, low concentrations of fatty acids have a positive impact on T cell proliferation and function, but high concentrations can lead to lipotoxicity-mediated apoptosis [127,128,129,130]. This phenomenon is not only dependent on the concentration but also on the type of fatty acid [127,128]. Furthermore, different T cell subsets react differently to fatty acid exposure.

CD8^+^ T cells are characterized by different metabolic requirements between effector and memory subsets. Compared to effector T cells (T_eff_), central-memory T cells (T_cm_) mostly rely on FAO fueled by de novo synthesis rather than by uptake of exogenous fatty acids. In this case, fatty acid synthesis is fueled by glucose uptake that serves as a substrate for triglyceride (TG) synthesis, which is then hydrolyzed in a process called lipolysis. The reason why T_cm_ rely on this futile cycle still needs to be elucidated [8,131]. However, it has been proposed in other cell types that this phenomenon may protect cells from lipotoxicity given the reduced FA uptake and the increased storage of FAs as TG [132]. Furthermore, the uncoupling of oxidative phosphorylation (OXPHOS) and ATP production is proposed as another option to maintain mitochondrial integrity during high lipid burden [82,133]. This hypothesis is supported by the fact that palmitate treatment enhances the expression of one of the mitochondrial uncoupling proteins, UCP2, in rat pancreatic islets [134]. Moreover, UCP2 inhibition enhances FFA-induced oxidative stress in human hepatocytes [135]. Interestingly, antigen-induced UCP2 expression in CD8^+^ T cells reduces glycolysis, FAS, and ROS production. Inhibition of UCP2 drives CD8^+^ T cell differentiation towards terminal short-lived effector T cells. Therefore, it is discussed as an interesting target for metabolic reprogramming of ex vivo expanded T cells for adoptive therapy [136,137]. Furthermore, increased expression of UCP2 might help T cells to overcome the detrimental effects of excess lipid levels regarding oxidative stress and mitochondrial dysfunction. Additionally, it is reported for CD8^+^ T cells that the ER stress-XBP1 pathway is activated by high cholesterol levels in the TME of a melanoma mouse model and is required for cholesterol-induced T cell exhaustion [138].

Regarding CD4^+^ T cells, it is believed that cell fate is likely determined by a combination of environmental cues, such as substrate availability, and cell-intrinsic programs, including the expression of certain transcription factors. Similar to CD8^+^ T cells, CD4^+^ effector and regulatory T cells (Tregs) have distinct metabolic profiles. Whereas T_eff_ mostly rely on glucose uptake and glycolysis, Tregs utilize FAO [139,140]. For instance, Tregs cannot be formed in the absence of FA or upon FAO inhibition. Given that T_eff_ rely less on FA, it would be expected that they are more susceptible to lipotoxicity in an environment of lipid overload. Interestingly, it was shown in a study that upon palmitate treatment, T_eff_ underwent lipotoxicity-mediated apoptosis, whereas Tregs were enriched, suggesting that Tregs are protected from lipotoxicity [132]. Notably, it seems that Foxp3 expression is sufficient to provide this advantage and induce metabolic reprograming based on FA utilization. In the aforementioned study, Foxp3 induced the upregulation of proteins involved in FAO and mitochondrial OXPHOS. Furthermore, the survival advantage of Foxp3-expressing T cells upon treatment with FAs seemed to be dependent on FAO since inhibition of enzymes from this pathway reversed the protective effects. Whether CD4^+^ T cells differentiate into T_eff_ or Tregs under identical microenvironmental conditions still needs to be fully understood. One possible explanation is the expression of pyruvate dehydrogenase kinase (PDHK), which favors glycolysis. CD4^+^ T_eff_ expresses higher levels of this enzyme, and PDHK inhibition leads to the enrichment of Tregs [141].

In summary, distinct metabolic programs determine how T cells behave in the lipid-rich TME. Given that T_eff_ mostly depend on glycolysis, and CD8^+^ memory T cells lack enzymes to catabolize or store very long chain FAs, TILs very frequently suffer from lipotoxicity caused by failed processing and accumulation of long-chain FAs in the cytoplasm. In addition, the lipid-rich TME affects not only T cell fitness and function but also T cell infiltration at the tumor site. For example, it was shown in a large-scale study that in tumors with a higher lipid metabolism, TIL frequencies positively correlate with a lower abundance of lipids in the TME [142]. Human and murine pancreatic ductal cancer, which are characterized by lipid enrichment, also display scarce infiltration with CD8^+^ T cells. In addition, these TILs are functionally impaired and exhausted due to high intracellular levels of lipids acquired from the lipid-rich TME [6], suggesting a detrimental effect of lipid accumulation in T cell infiltration and trafficking to solid tumors. Interestingly, the lipotoxicity-mediated poor migration and T cell dysfunction could be a potential mechanism to explain the lack of success of immunotherapies, especially against solid tumors. Moreover, the lack of TILs and their already exhausted phenotype, as well as the survival advantage for immunosuppressive Tregs in such a lipid-enriched environment [6,138,143,144], carries the risk of failure of adoptive T cell therapy when ex vivo-expanded TILs are used. Besides, systemic hyperlipidemia could also drive exhaustion of adoptively transferred T cells and decrease their anti-tumor efficiency. To our knowledge, this has not yet been addressed in murine or clinical studies but could be an interesting target for improving adoptive T cell therapy.

### 4.2. The Role of CD36 Expression and Ferroptosis in T Cells

One of the key molecules responsible for lipid uptake is the FA transporter CD36, which is expressed in many cell types and tissues, including T cells. Therefore, many studies have focused on the relationship between the enhanced presence of lipids in the TME, CD36 expression and the subsequent effects of FAs, such as lipid peroxidation-mediated ferroptosis.

Again, the effects of CD36 expression in different T cell subsets are accompanied by different outcomes. Wang et al. reported an increased expression of CD36 and PPAR-β in intra-tumoral Tregs, which promoted Treg survival and progression in their murine cancer model [145]. This further strengthens the point of superior protection of Tregs against lipotoxicity. Contrarily, CD36 expression and increased uptake of lipids by CD8^+^ T cells are associated with impaired effector function and proliferation as well as increased exhaustion [138,143]. Interestingly, TILs had higher CD36 expression than CD8^+^ T cells from healthy adjacent tissue, and TILs with an exhausted phenotype (PD1^+^ TOX^+^) had an even higher CD36 expression compared to their non-exhausted counterparts (PD1^+^ TOX^−^) [143]. In addition, TILs from long-survival patients displayed overall reduced expression of CD36 as compared to patients with a worse prognosis. CD36 was not only associated with overall survival but also with response to checkpoint inhibitor therapy since CD36 expression was downregulated in patients responding to anti-PD-1 treatment [144]. Similarly, in a different study, TILs from melanoma, colon adenocarcinoma and liver tumors also displayed enhanced CD36 expression [146,147]. This increased CD36 expression is the result of a TME rich in cholesterol, FAs, and oxidized lipids, as shown in these studies. For instance, in cyclodextrin-treated tumors, which are depleted from cholesterol, CD36 expression is reduced in comparison to untreated conditions [144].

Besides T cell function and exhaustion, CD36 has also been linked to ferroptosis. Ferroptosis is caused by lipid peroxidation, which requires lipids, iron, and ROS. Interestingly, when analyzing TILs from CD36^−/−^ and WT mice, two pathways were differentially expressed: the sirtuin pathway, which was activated, and OXPHOS, which was downregulated in CD36^−/−^ T cells. This is beneficial since OXPHOS is a major ROS producer, and the sirtuin pathway protects from oxidative stress. Therefore, CD36^high^ expressing T cells are characterized by increased lipids and ROS levels. Moreover, CD36^−/−^ T cells also displayed reduced expression of genes involved in ferroptosis and lipid peroxidation. In addition to reduced expression of these key genes, CD36^−/−^ T cells indeed displayed reduced lipid peroxidation, reduced iron levels and reduced cell death [144]. Consequently, it seems that CD36 expression, induced by the lipid-rich TME, is responsible for increased lipid peroxidation and, subsequently, ferroptosis.

The same authors later published another study focusing on the metabolic differences between different CD8^+^ T cell subsets. In this study, they reported that both murine and human cytotoxic T lymphocytes subset 9 (Tc9), which are activated by IL-9 signaling, display lower lipid peroxidation and ferroptosis levels than Tc1 cells. This reduction was shown to be due to activated STAT3 signaling, increased FAO, and mitochondrial activity. These results supported that Tc9 cells are more suitable for adoptive transfer since they adapt better to lipid-rich TMEs. Interestingly, they also showed that TILs in melanoma patients expressed lower IL-9 levels than circulating T cells, which was correlated with higher peroxidation and ferroptosis rates [148]. These findings indicate an advantage of using T cells with an active lipid catabolism for adoptive immunotherapies.

Similar results were observed with oxidized LDL (oxLDL) in murine T cells. Uptake of oxLDL via CD36 led to lipid peroxidation, decreased cytokine production and increased expression of exhaustion markers. In contrast, CD36^−/−^ T cells displayed lower levels of lipid peroxidation, oxidative stress, and p38, which is responsible for inducing cell death in CD8^+^ T cells. These results, including the alterations in viability and T cell function, could be reverted by GPX4 overexpression, supporting the idea that CD36 expression leads to T cell dysfunction via ferroptosis [143]. Moreover, it has been shown that in the absence of GPX4, T cells undergo apoptosis due to enhanced lipid peroxidation, and GPX4 seems to be essential to protect T cells against viral and parasitic infections, further stretching the importance of this pathway for T cell function and persistence [149,150].

In conclusion, several studies point out the implications of ferroptosis as well as the important role of CD36 in T cell composition, fitness and function in the context of lipid-rich TMEs, suggesting that CD36 expression in T cells used for adoptive cell transfer might be a limiting factor for the therapy outcome.

## 5. Conclusions and Perspectives

In summary, adoptive T cell therapy provides a promising treatment strategy leading to durable responses. However, this therapeutic option is not applicable to every tumor entity, and not every patient reaches favorable results. Besides low expression of tumor (neo)antigens and expression of inhibitory receptors, a hostile TME further limits the efficacy after T cell transfer. Amongst others, enrichment of lipids in the TME can be found in many tumor entities. This accumulation can result either from systemic causes, such as obesity and aging, or from reprogramming of the stromal compartment by the tumor cells themselves to meet their need for nutrients and bioenergetic intermediates. Due to their high metabolic flexibility and development of apoptosis resistance, most cancer cells are protected from this lipotoxicity. However, immune cells (with anti-tumor activity) are often incapable of coping with this high lipid burden, resulting in loss of effector function, exhaustion, and cell death, summarized as lipotoxicity. Interestingly, the accumulation of lipids in the TME has opposite effects on T cells depending on the subset. On the one hand, a lipid-enriched TME promotes the survival of Tregs due to their metabolic reprogramming towards FA utilization and FAO. Importantly, enhanced survival of Tregs in the TME leads to the suppression of several innate and adaptive effector immune cells, cumulating in mitigated anti-tumor immunity. On the other hand, tumor-specific effector T cells and consequently also transferred T cells are inhibited in their function and survival by lipotoxicity, which also mediates tumor growth. Therefore, it is important to understand and overcome the mechanisms of the lipid-mediated tumor immune evasion in order to develop novel approaches and improve current therapies.

Overcoming this “metabolic immunobarrier” could be achieved on three levels: (1) systemically, (2) at the level of tumor or (3) at the level of T cells. Targeting the systemic deregulated lipid metabolism by weight management in overweight and obese cancer patients is discussed to promote survival and delay progression [151]. Unfortunately, the data situation in human cancer patients is so far rather insufficient and requires further studies that are currently running worldwide [152,153,154,155]. Moreover, direct pharmacological regulation of lipid concentrations in the blood and TME during cancer therapy could also provide an advantage for patients’ outcomes (Table 1). For instance, metformin, also known as a fasting-mimicking substance, reduces blood cholesterol and triglycerides in hyperglycemic and diabetic patients. However, to date the use of metformin has not (yet) been proven beneficial in combination with anti-tumor therapies [156]. The use of statins, which reduces blood cholesterol by HMG-CoA reductase inhibition, was found to improve patients’ survival in several solid tumor entities, such as breast cancer, pancreatic cancer, gastric cancer and lung cancer [157]. Furthermore, statins also improved patients’ survival and delayed the need for chemotherapy in CLL [64,65,158,159], whereas others report a stage-dependent benefit of statin use [158]. Injection of simvastatin into the tumor bed of a B16 tumor mouse model reduced the cholesterol content in the tumor and the TME resulting in lower CD8^+^ T cell exhaustion [138]. Interestingly, the combinations of statins with chemotherapy or immunotherapy were and are already under investigation in clinical trials [157,160].

However, if these effects are seen due to the systemic lowering of lipids or due to direct inhibition in tumor cells and cells of the TME has to be determined. For instance, inhibition of proliferation and induction of apoptosis by statins as a single agent has been described for several cancer entities [157]. Amongst others, inhibition of the mevalonate pathway, promotion of ferroptosis, and regulation of autophagy in cancer cells are discussed as the underlying mechanism for these statin-mediated effects. Furthermore, the combination of statins with other anti-cancer drugs has been shown to increase the efficacy of treatments [157,160]. The combination of simvastatin with pentoxifylline, a sensitizer to chemotherapy and radiotherapy from the xanthine family, drives apoptosis, autophagy, and cell cycle arrest in triple-negative MDA-MB231 breast cancer cells [161]. In CLL, statins enhance the sensitivity to the proapoptotic BCL2-inhibitor venetoclax [162]. Moreover, pharmacologic inhibition of FAO in human leukemia cells can also reinforce apoptosis sensitivity by inhibition of BCL2, similar to statins [163]. Inhibition of the B cell receptor signaling on CLL cells by ibrutinib reduces LPL expression and FFA metabolism in vitro, interrupting energy production and survival [164]. In BRAF(V600E)-mutant residual therapy-resistant melanoma cells, inhibition of peroxisomal FAO overcomes resistance to BRAF/MEK inhibitors [165]. Furthermore, inhibition of lipid uptake in cancer cells provides an additional target for combination therapy. In this regard, CD36 expression is repeatedly discussed as a prognostic marker for several cancer entities concerning invasion and metastasis [166]. For instance, high expression of CD36 promotes growth, migration and tamoxifen resistance of breast cancer cells, which is reduced by CD36 siRNA [167]. Additionally, CD36 is essential for anti-HER2 therapy resistance and is associated with poor clinical prognosis [168]. CD36-expressing chronic myeloid leukemia cells are less sensitive to imatinib treatment [169]. Interestingly, a blocking anti-CD36-antibody is currently under development for cancer treatment [170].

As mentioned above, excessive uptake of lipids via CD36 by CD8^+^ T cells leads to ferroptosis, and CD36 blocking/knockout prevents T cell lipotoxicity [143,144]. Therefore, adoptive T cell transfer might also benefit from CD36 inhibition/knock-down. Moreover, targeting CD36 by, e.g., an anti-CD36 antibody, could decrease one important immunosuppressive compartment in tumor tissues because Tregs upregulate CD36 in the TME [171]. Thus, CD36 inhibition could have a dual effect on the T cell compartment in the TME: (1) preserving effector T cells and (2) reducing immunosuppressive Tregs. Furthermore, increasing fatty acid catabolism and oxidative metabolism in TILs to cope with excess intracellular FFAs, for instance, by enhancing PPARα signaling with fibrates, could also improve effector function and survival in a lipid-enriched TME [172,173]. Recently, Kim et al. published an interesting approach to metabolically reprogram T cells in B16F10 tumor-bearing mice. They encapsulated fenofibrate in amphiphilic polygamma glutamic acid-based nanoparticles carrying an anti-CD3e f(ab′)2 fragment on the surface that are taken up by T cells. Injection of these nanoparticles into tumor sites increased FAO and restored mitochondrial function of TILs, resulting in effective anti-tumor function [174]. Treatment with these nanoparticles prior to adoptive transfer could enable T cells to overcome a low-glucose but lipid-rich TME and break the metabolic barrier, especially in solid tumors.

**Table 1 biomolecules-12-01182-t001:** Selected examples of intervention in fatty acid metabolism for cancer therapy.

	Drug	Tumor Entity	Target	Effect	Source
Clinical studies	Metformin (monotherapy/combination therapy)	Various human tumor entities	Inhibition of hepatic gluconeogenesis and lipid synthesis Decreased adipose tissue fatty acid synthesis and lipolysis Decreased pancreatic insulin secretion Increased muscle glucose uptake	No beneficial effects proven	[156]
	Statins	Various human tumor entities	Inhibition of HMG-CoA reductase	Improvement of patients’ survival in several tumor entities	[157,160]
		CLL	Inhibition of HMG-CoA reductase	Improvement of patients’ survival and outcome;	[64,65,158,159]
				delayed need for chemotherapy	[64]
	Fibrates + immune checkpoint blockade	Non-small cell lung cancer (NSCLC)	Peroxisome proliferator-activated receptor agonist	Improved overall survival	[172]
In vivo	Thioridazine + BRAF/MEK inhibitors	Human melanoma-bearing mice	Peroxisomal FAO inhibitor	Increased sensitivity to BRAF/MEK inhibitors in melanoma persister cells	[165]
	Simvastatin	B16 tumor mouse model	Inhibition of HMG-CoA reductase	Lower CD8^+^ T cell exhaustion	[138]
	Nanoparticles carrying fenofibrate and surface anti-CD3e f(ab)2 fragment	B16 tumor mouse model	Fatty acid metabolism of TILs	Increased FAO and mitochondrial functions of TILs; improved T cell anti-tumor function	[174]
	Ferrostatin-1	B16 tumor mouse model	Pretreatment of T cells with ferroptosis inhibitor before adoptive transfer	Improved T cell anti-tumor function	[144]
	Fenofibrate (+PD-1 blockade)	Mouse melanoma models	Fatty acid metabolism of TILs	Improved T cell anti-tumor function	[173]
In vitro	Simvastatin + pentoxifylline	Triple-negative MDA-MB231 breast cancer cells	Inhibition of HMG-CoA reductase	Apoptosis, autophagy, and cell cycle arrest in cancer cells	[161]
	Statin + venetoclax	CLL	Inhibition of HMG-CoA reductase	Enhanced the sensitivity to venetoclax	[162]
	Etomoxir/ranolazine + ABT-737 (Bcl-2/Bcl-xL inhibitor)	Human leukemic cells	Inhibition of FAO	Increased apoptosis sensitivity in leukemic cells	[163]
	Ibrutinib	CLL	B cell receptor inhibition	Reduction of LPL expression and FFA metabolism in CLL	[164]
	CD36-blocking antibody	Multiple melanoma	Fatty acid metabolism of TILs	Reduction of ferroptosis in T cells	[144]

Consequently, there are several options to reduce potential lipotoxicity in T cells for adoptive therapy (Figure 5). Lifestyle modifications could help on a systemic level not only to reduce the risk of tumor development but also to improve therapeutic outcomes by monitoring weight and increasing physical activity when possible, resulting in normal blood lipid levels and less lipid storage in adipose tissues. However, further carefully designed studies are needed to extend the knowledge on the influence of overweight/obesity as well as weight management on cancer development and therapy. Drugs used for systemic treatment of dyslipidemia, e.g., statins, have shown promising results as mono- and combinational therapy with other anti-cancer drugs (Table 1). However, these promising results are possibly also effects of the drugs on tumor cells themselves. Direct targeting of lipid metabolism and uptake in cancer cells and T cells are two other intervention options (Table 1). Inhibition of lipid uptake on the tumor side, e.g., using anti-CD36 antibodies, could inhibit tumor progression and metastasis on the one hand and promote anti-tumor T_eff_ without supporting Tregs on the other hand due to reduced lipid content producing lipotoxicity. Furthermore, deletion of CD36 in T cells for adoptive transfer might also produce favorable outcomes. Inhibition of lipid metabolism and thus promotion of lipotoxicity in tumor cells and, conversely, promotion of lipid catabolism in transferred T cells to overcome the lipid burden need to be further explored in T cell-based therapies and may prove to be a good way to improve treatment outcomes.

## Figures and Tables

**Figure 1 biomolecules-12-01182-f001:**
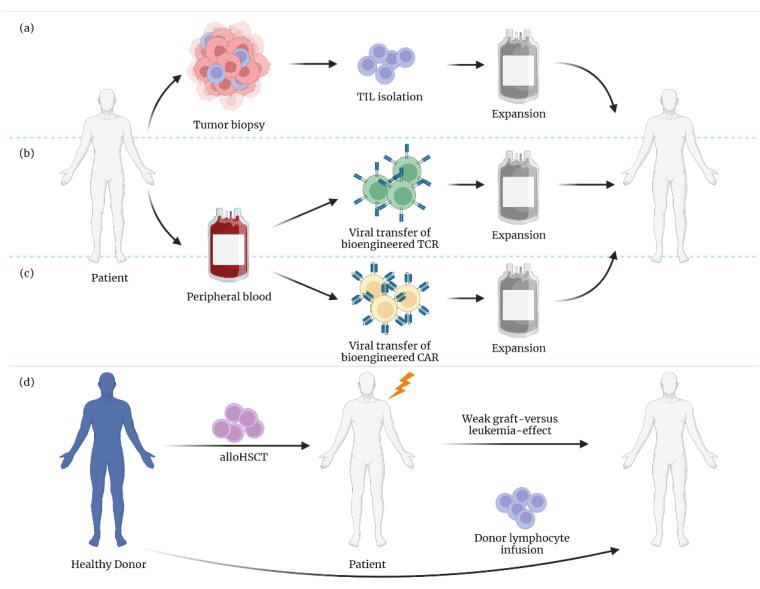
Adoptive T cell transfer. (**a**) Tumor-infiltrating lymphocytes are isolated from tumor tissue and ex vivo expanded. Afterwards, expanded cells are delivered back to the patient. (**b**,**c**) Peripheral blood is collected from cancer patients. T cells are then isolated and transduced with either (**b**) a T cell receptor (TCR) specific against tumor neoantigens or (**c**) a chimeric antigen receptor (CAR) targeted against cancer-related antigens. Both types of T cells are then expanded in vitro before infusion into the patient. (**d**) In allogeneic hematopoietic stem cell transplantation (alloHSCT), the patient’s immune system is eliminated by conditioning and replaced by foreign immune cells arising from healthy donor hematopoietic stem cells. However, leukemic cells can persist or reoccur due to a weak graft-versus-leukemia effect. Infusion of allogeneic peripheral T cells from the stem cell donor after transplantation, termed donor lymphocyte infusion (DLI), can eradicate residual leukemic cells.

**Figure 2 biomolecules-12-01182-f002:**
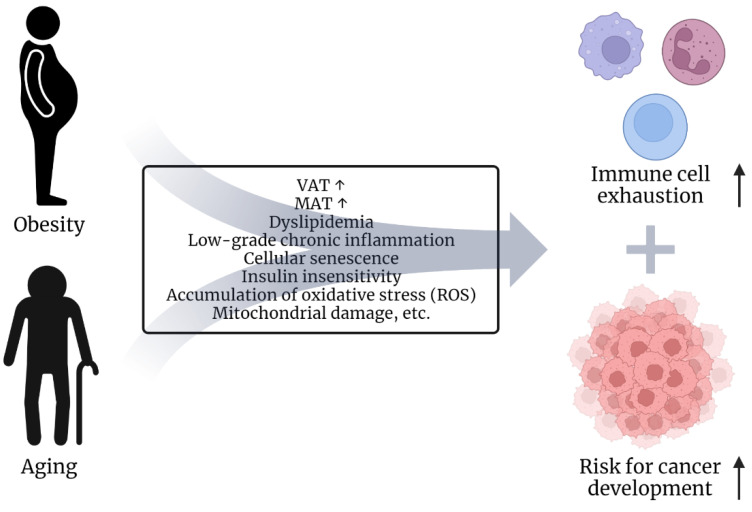
Role of obesity and aging in oncogenesis. Obesity and aging share characteristics. In both conditions, an increase in visceral and bone marrow adipose tissue is observed [37,38,40]. In addition, dyslipidemia [39], low-grade chronic inflammation leading to cellular senescence [31,46], as well as increasing insulin insensitivity [46], accumulation of ROS [46], mitochondrial damage [46] etc., are considered mutual concomitants of obesity and aging. These factors may eventually lead to immune cell exhaustion/dysfunction and an increased risk of developing cancer over time.

**Figure 3 biomolecules-12-01182-f003:**
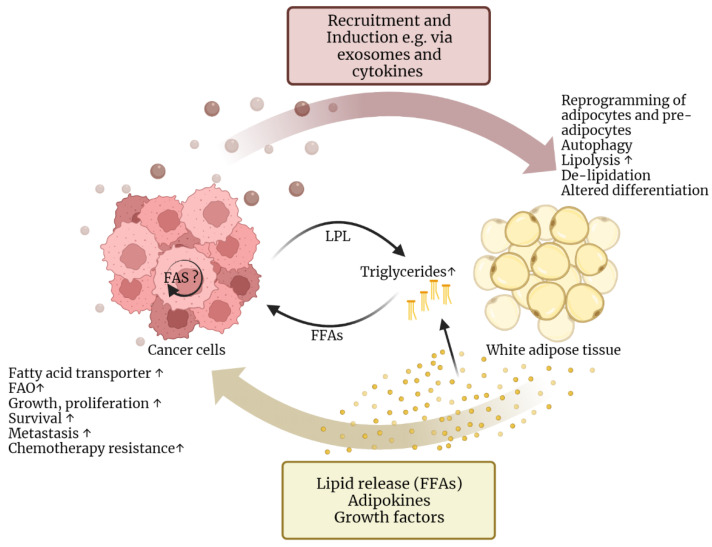
Reciprocal reprogramming of cancer cells and adipocytes. Adipocytes are part of the tumor microenvironment in multiple cancer entities. They can be induced and activated by tumor-derived factors, such as exosomes and cytokines. This results in the reprogramming of adipocytes and adipose stem cells, which includes autophagy, lipolysis, delipidation due to the release of free fatty acids (FFAs), de-differentiation of mature adipocytes and altered differentiation of adipose stem cells into senescent cancer-associated adipocytes. In addition to FFAs, adipocytes also secrete tumor-promoting adipokines and growth factors. FFAs can be directly taken up by the tumor cells or increase triglycerides (TGs) in the blood. These TGs can be hydrolyzed by tumor-derived LPL to FFAs, which then are taken up by tumor cells. In turn, uptake of FFAs from the microenvironment increases the expression of fatty acid transporters and FAO, stimulates growth and proliferation, increases survival, the ability for metastasis and resistance to chemotherapy. Cancer cells can also upregulate intrinsic fatty acid synthesis (FAS). However, newly synthesized FAs are not secreted and are rather used as substrates in anabolic processes and signaling.

**Figure 4 biomolecules-12-01182-f004:**
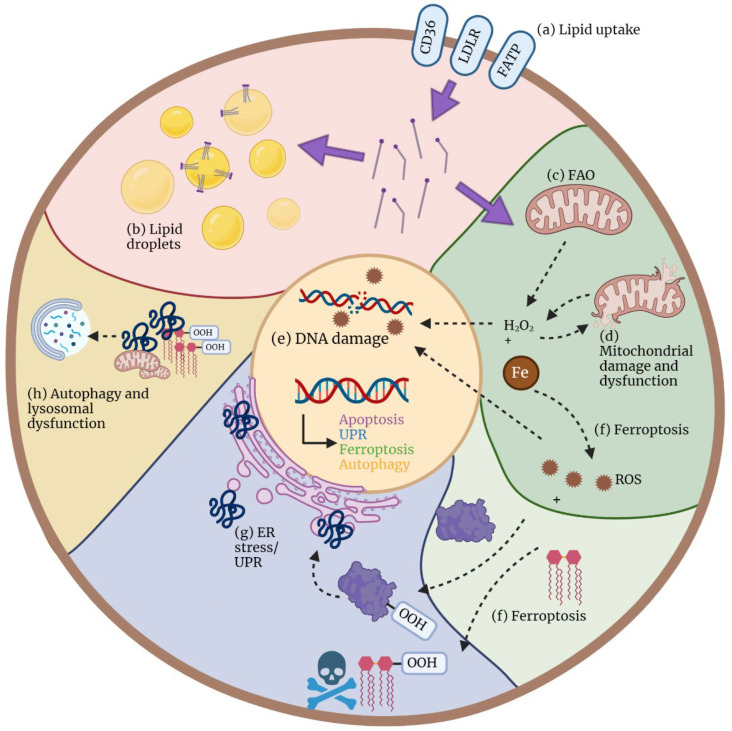
Mechanisms of lipotoxicity. Excessive lipids, especially free fatty acids (FFAs), are frequently present in the TME. (**a**) FFAs can be taken up into the cell via specific transporters, such as CD36, LDL receptor (LDLR) and fatty acid transporter proteins (FATPs). Once in the cytoplasm, FFAs have different fates and might cause lipotoxicity via different mechanisms, which are often interconnected. (**b**) To prevent lipotoxicity, FFA can be stored in lipid droplets as triglycerides (TGs) for later use or simply as a protective mechanism (pink). Excess FFAs in the cytoplasm trigger the following effects: (**c**) FFAs can be transported into the mitochondria to fuel FAO and, subsequently, oxidative phosphorylation (OXPHOS). One of the products of these reactions is hydrogen peroxide (H_2_O_2_), one of the most abundant reactive oxygen species (ROS) that can lead to oxidative stress, mitochondrial (**d**) and DNA damage (**e**), and, subsequently, cell death. (**f**) Excessive ROS in the presence of iron (Fe) can lead to lipid and protein peroxidation and, subsequently, a recently described type of regulated cell death termed ferroptosis. (**g**) Protein peroxidation, among other processes, leads to the accumulation of misfolded proteins in the endoplasmic reticulum (ER), which causes ER stress and triggers the unfolded protein response (UPR). (**h**) Misfolded proteins, oxidized lipids, and malfunctional organelles are eliminated in the lysosomes via the process of autophagy, whose overload results in lysosomal dysfunction.

**Figure 5 biomolecules-12-01182-f005:**
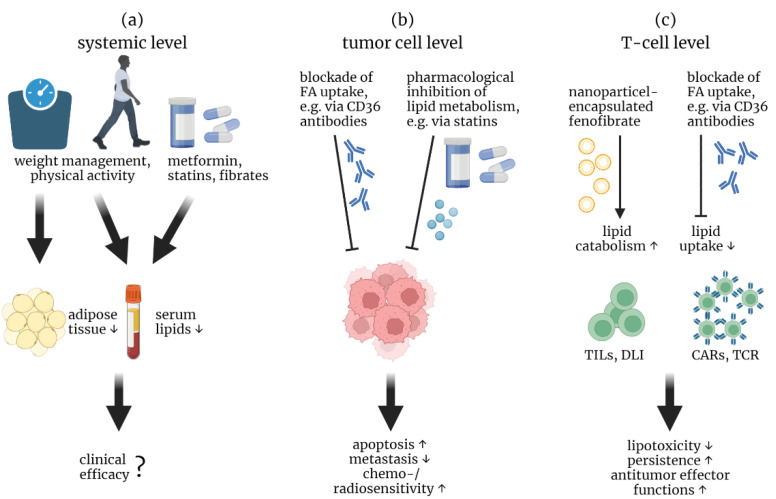
Therapeutic approaches to circumvent lipotoxic barriers. Overcoming T cell-related lipotoxicity in cancer for adoptive therapy could be achieved at three levels: (**a**) on a systemic level, (**b**) on a tumor cell level and (**c**) on a T cell level. (**a**) To decrease tumor-supporting adipose tissue and consequently lower serum lipids in overweight and obese patients, weight management and physical activity could be beneficial. In addition, metformin, statins and fibrates could be used to reduce serum lipids. However, the clinical efficacy and reliability still have to be demonstrated. (**b**) Blockade of lipid uptake, e.g., by an anti-CD36 antibody, and pharmacological inhibition of lipid metabolism, e.g., by statins, restrict the lipid pool in tumor cells from which the cells could generate energy and molecules for biosynthesis. As a result, tumor cells undergo apoptosis, exhibit reduced metastasis and invasiveness, and become more sensitive to chemotherapy and radiotherapy. (**c**) Enhancing lipid catabolism, e.g., by fenofibrate in CD3-targeted nanoparticles, and blocking the uptake of excess lipids could prevent lipotoxicity in TILs and transferred T cells in a lipid-enriched TME, resulting in T cell persistence and sustained anti-tumor effector function.

## Data Availability

Not applicable.

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
