# Peer review of "Lipotoxicity as a Barrier for T Cell-Based Therapies"

_biomolecules, 2022, doi:10.3390/biom12091182_

Round 1

Reviewer 1 Report

Review Report Biomolecules 1865121

This is a Review article on the role of lipids, fatty acid metabolism and lipotoxicity as a barrier for T cell-based therapies. It focuses on cancer and the limitations caused by deregulated lipid metabolism within the tumor microenvironment (TME). The manuscript is organized in five sections. In the first section (Introduction) the authors provide the background for adoptive cell-based therapies, the TME and the major barriers it creates, and also introduce the main lipid categories and their roles in cellular and systemic homeostasis, implications in pathological conditions and introduction to the term “lipotoxicity”. In the second section (Sources of lipids within the tumor microenvironment), the authors discuss obesity, ageing and cancer under the common denominator of deregulated lipid metabolism and effects on T cell responses. In the third section (Mechanisms of cellular lipotoxicity), the authors dive into the molecular understanding of lipotoxicity and specific mechanisms by which lipids can have detrimental effects to cellular functions, including the role of CD36, LDLR and FATPs in lipid transport, ER stress, unfolded protein response, ROS production, oxidative stress, ferroptosis, macromolecular damage, autophagy and lysosomal dysfunction. In the fourth section (Implication of lipotoxicity on T cell-based therapies), the authors discuss the specific and differential effects of fatty acids on T cell subsets such as CD8 and CD4 effector as well as regulatory T cells. Finally, in the fifth section (Conclusion and perspectives), the authors thoroughly discuss the potential to overcome the TME “immunobarrier” by intervening with lipid metabolism at three levels: 1) systemically, 2) at the tumor level and 3) at the T cell level. This is a high quality, comprehensive, thorough, novel, timely and up to date review article. It includes five figures which efficiently reflect the text information. The authors integrated past and latest literature and discussed gaps in knowledge, controversies and future perspectives.

There are only minor corrections for improving the text:

Line 153-154: Suggested modification: … survival of tumor cell cumulates leads to glucose depletion and hypoxia.

Line 214: Just a space missing: …TME [71,72].

Line 293: Just a space missing: … [83]. FFAs…

Line 295: Just a space missing: … [84]. Conclusively,…

Line 295: … surplus of fatty acids…

Line 369: …cristae.

Line 483: …adoptively transferred…

Line 530: …IL-9…

Author Response

There are only minor corrections for improving the text:

 Line 153-154: Suggested modification: … survival of tumor cell cumulates leads to glucose depletion and hypoxia.

Line 214: Just a space missing: …TME [71,72].

Line 293: Just a space missing: … [83]. FFAs…

Line 295: Just a space missing: … [84]. Conclusively,…

Line 295: … surplus of fatty acids…

Line 369: …cristae.

Line 483: …adoptively transferred…

Line 530: …IL-9…

Thank you for your kind and thorough revision of our manuscript. We adjusted the text accordingly.

Reviewer 2 Report

The authors review the effect of excessive exposure of T-cells to fatty acids, with implications on T-cell based anticancer strategies. This is a well-structured manuscript that describe the signaling and mechanism of action of lipid accumulation in tumoral microenvironment. 

Some minor comments for improving this manuscript are:

- Review the abbreviations in the text. Some not have the full name (e.g. OXPHOS).

- A Table that summarizes the molecular effects produced by lipotoxicity and the evidence of drugs used may be useful for the readers.

Author Response

Thank you for your time and effort for the review of our manuscript.

Review the abbreviations in the text. Some not have the full name (e.g. OXPHOS).

Thank you for pointing this out. We have thoroughly checked the manuscript again to make sure that every abbreviation appears as the full term first. In case of OXPHOS, we have added the full name in line 433:

“Furthermore, uncoupling of oxidative phosphorylation (OXPHOS) and ATP production …”

A Table that summarizes the molecular effects produced by lipotoxicity and the evidence of drugs used may be useful for the readers.

Thank you for your comment. We have accordingly summarized possible drugs that are used in the context of fatty acid metabolism and cancer therapy as shown in the new Table 1. However, there are currently no treatments that specifically address molecular effects of (T-cell) lipotoxicity and the possibilities for cancer therapy are outlined in our Conclusion and perspectives. We hope for the reviewer’s understanding.